# Lipidomics—Paving the Road towards Better Insight and Precision Medicine in Rare Metabolic Diseases

**DOI:** 10.3390/ijms24021709

**Published:** 2023-01-15

**Authors:** Martina Zandl-Lang, Barbara Plecko, Harald Köfeler

**Affiliations:** 1Division of General Pediatrics, Department of Pediatrics and Adolescent Medicine, Medical University of Graz, 8036 Graz, Austria; 2Core Facility Mass Spectrometry, ZMF, Medical University of Graz, 8036 Graz, Austria

**Keywords:** lipidomics, mass spectrometry, rare diseases, orphan diseases

## Abstract

Even though the application of Next-Generation Sequencing (NGS) has significantly facilitated the identification of disease-associated mutations, the diagnostic rate of rare diseases is still below 50%. This causes a diagnostic odyssey and prevents specific treatment, as well as genetic counseling for further family planning. Increasing the diagnostic rate and reducing the time to diagnosis in children with unclear disease are crucial for a better patient outcome and improvement of quality of life. In many cases, NGS reveals variants of unknown significance (VUS) that need further investigations. The delineation of novel (lipid) biomarkers is not only crucial to prove the pathogenicity of VUS, but provides surrogate parameters for the monitoring of disease progression and therapeutic interventions. Lipids are essential organic compounds in living organisms, serving as building blocks for cellular membranes, energy storage and signaling molecules. Among other disorders, an imbalance in lipid homeostasis can lead to chronic inflammation, vascular dysfunction and neurodegenerative diseases. Therefore, analyzing lipids in biological samples provides great insight into the underlying functional role of lipids in healthy and disease statuses. The method of choice for lipid analysis and/or huge assemblies of lipids (=lipidome) is mass spectrometry due to its high sensitivity and specificity. Due to the inherent chemical complexity of the lipidome and the consequent challenges associated with analyzing it, progress in the field of lipidomics has lagged behind other omics disciplines. However, compared to the previous decade, the output of publications on lipidomics has increased more than 17-fold within the last decade and has, therefore, become one of the fastest-growing research fields. Combining multiple omics approaches will provide a unique and efficient tool for determining pathogenicity of VUS at the functional level, and thereby identifying rare, as well as novel, genetic disorders by molecular techniques and biochemical analyses.

## 1. Introduction

According to the European Union, rare diseases, also known as orphan diseases, are defined by a prevalence of one in 2000, affecting approximately 30 million people in the EU and 400 million patients worldwide. To date, 6172 distinct entities of rare diseases have been described, with approximately ~72% of genetic origin and ~70% of exclusively pediatric onset [1]. Due to the rarity of single disease entities, diagnosis is hampered by a lack of recognizable phenotypes and limited clinical knowledge/expertise. Even within specific disease groups, the genetic background can be very heterogeneous. The resulting consequences are a lack of treatment and that three out of ten children will not live through their fifth birthday.

Inborn errors of metabolism are rare disorders that are characterized by the accumulation or deficiency of small or complex molecules, resulting in acute and chronic organ dysfunction [2]. Omics approaches, including genomics, metabolomics and proteomics, try to combine information on metabolite patterns, altered protein expression and genetic variants to proof the pathogenicity of mutations affecting metabolic pathways on a functional level. Next generation sequencing (NGS) of all 22.000 human genes (genomics) enables an accurate diagnosis in only 30–40% of cases with presumable inherited etiology, while the remainder is either left with variants of unknown significance (VUS) in known disease genes or, even more difficult, in candidate genes that await proof of their clear association with a disease [3]. A combined approach of NGS and other omics techniques can thus help to establish the diagnosis of ultra-rare genetic disorders. The delineation of novel biomarkers is not only crucial to prove the pathogenicity of VUS, but provides surrogate parameters for monitoring disease progression and therapeutic interventions. Metabolic biomarkers, if present, do enhance the diagnostic yield of NGS to 50–90%, which underlines the need for biomarker identification at a broader level [4,5]. Since metabolic disorders are amenable to special treatment, fast diagnosis is a prerequisite for improved patient outcomes and the prevention of irreversible damage. In addition, the identification of novel lipid biomarkers is essential for neurological diseases due to the high impact of lipid metabolism in the central nervous system (CNS).

## 2. Lipids and Pathologies

Lipids are defined as hydrophobic or amphipathic small molecules with a high solubility in organic solvents. Following water, lipids are the second most abundant components in mammalian cells. The lipidome comprises tens of thousands of different species, which are broadly subdivided into simple lipids, e.g., fatty acids (FA), or complex lipids, e.g. sphingolipids (SL), acylglycerols or phospholipids (PL). Lipids are crucial for structural compartmentalization by being major constituents of the semi-permeable plasma membranes formed by a lipid bilayer, majorly composed of PL and proteins. Lipid rafts are particularly rigid regions in the membrane characterized by condensed packing of cholesterol (Chol), SL and glycolipids, including acyl chains with a high degree of saturation, which are less fluid than the PL-rich liquid-disordered plasma membrane [6]. The formed microdomains are important areas for cell function and are able to regulate the conformation and function of many GPI-anchored proteins, as well as proteins involved in neurodegeneration such as prions and amyloid precursor protein (APP). Most lipids show heritabilities of 50% or more, control the biophysical property of the membrane and the lipid composition can change in response to pathophysiological modifications [7]. As such, lipid domains are crucial and serve as a point of entry for viruses infecting human cells, such as HIV and Ebola [8].

In addition to its barrier function, lipids play a crucial role in energy storage and signal transduction. These so-called “endogenous bioactive lipids” serve as key pathophysiological mediators, regulating several intracellular functions, including apoptosis, proliferation, response to stress and inflammation [9,10,11]. Especially, SL are known for their contribution in inflammatory processes by controlling intracellular trafficking and signaling, cell growth, adhesion, vascularization, survival and apoptosis [12]. Located within myelin sheaths, SL play a fundamental role in neuronal function and insulation. Signaling SLs, such as sphingosine-1-phosphate (S1P) and ceramide-1-phosphate (C1P), are defined as bioactive lipid hormones that function in various physiological processes [13,14]. S1P is reduced in the brains of patients suffering from stroke or multiple sclerosis (MS) [15]. Ceramides serve as building blocks for complex SL, such as Spingomyelin (SM) and glycosylceramides. Ceramides mediate pro-inflammatory processes and cytokine production, but are also important for endothelial and vascular integrity [16]. Changes in SL, such as SM and ceramides, are associated with atherogenesis and vascular dysfunction [17]. Glycohydrolases and sphingomyelinases are required for the degradation of complex SL. Defects in these specific catabolic enzymes lead to the manifestation of sphingolipidoses, such as Niemann Pick, Gaucher, Fabry, Farbers and Tay-Sachs disease [12]. Gangliosides belong to the glycosphingolipids group and are composed of ceramides and oligosaccharides with one or more sialic acids. These lipids are localized at the surface of cell membranes (e.g., lipid rafts, vesicles and synapses) associated with intercellular signaling, as well as with various diseases especially affecting neurodegeneration [16].

Even though over one hundred genetic disorders are related to defects of lipid synthesis and remodeling [18], our knowledge regarding the connection of genetic pathways and the quantitation of lipids is still very limited (Figure 1). It has been shown that, particularly disorders of the CNS and peripheral nervous system (PNS), are connected with dyslipidemia. The human brain is defined as the lipid-richest organ of the body, with a unique lipid turnover rate compared to the rest of the body. Lipids in the brain are crucial for synaptogenesis. The major lipid classes in the brain are Chol, SL and PL, including phosphatidylcholine (PC), phosphatidylethanolamine (PE) and phosphatidylserine (PS) as the three most abundant species [19]. The lipid composition is unique, depending on individual cell type and age. Due to the maturing of membrane and myelin sheaths in the brain, PC is the predominant lipid class at birth (with 50% of the overall lipid content), but then declines, whereas the SM content increases from 2% to 15% at 3 years of age [6]. Myelin reveals a lipid content of approximately 80%; white and gray matter of approximately 55% and 38%, respectively. Ether-lipids, such as plasmalogens, are ubiquitously present in the human body, but are particularly important in the heart, spleen and brain. Plasmalogens account for approximately 20% of PL in the adult brain, are highly enriched in polyunsaturated FA and constitute essential compounds in myelin formation [20,21]. Low levels of plasmalogens are associated with brain abnormalities such as hypomyelination, microglia neuroinflammation, defects in neuronal migration and cerebellum dysformation [22,23,24,25,26].

A defective lipid metabolism can affect a variety of organs and has been linked to several growing groups of orphan diseases. Disorders of complex lipid synthesis are responsible for prominent motor manifestations due to upper and lower motor neuron degeneration [18,27]. To date, primary dyslipidemia is associated with at least 49 neurological disorders, including ataxia, epilepsy, movement disorders, neuropathy and hereditary spastic paraparesis (Figure 1).

In addition, 47 ophthalmologic gene defects have been associated with the lipid metabolism, including Conradi Hunermann and Smith Lemli Optiz syndromes [18,27] (Figure 1). So far, lipid synthesis and remodeling have been linked to approximately 30 inherited diseases with dermatological manifestations, such as ichtyosis and neuro-cutaneous syndromes, including Sjögren-Larsson syndrome. Approximately 20 inherited orthopedic diseases, such as growth disorders or dysplasia, have been associated with a defective lipid metabolism. Since the liver is the central organ for FA metabolism, a minimum of 20 hepatic gene defects, such as jaundice, hepatosplenomegaly and hepatic steatosis, including ultrarare disorders such as Wolman disease, have been linked to dyslipidemia. In addition, complex lipid synthesis and remodeling disorders are known to be involved in approximately 20 defects of the skeletal and cardiac muscle, including myalgia, myopathy and primary cardiac presentations (Figure 1). Dyslipidemia is also linked to several other congenital disorders, such as diseases of the kidney and the immune system, and, presumably, numerous genetic defects which are yet to be discovered.

## 3. Lipidomics

The metabolome represents the global spectrum of low molecular-weight (<1 kDa) metabolites in a distinct biological sample, and provides temporal information about its biochemical environment [28]. As such, alterations IN the metabolome reflect genomic, transcriptomic and proteomic changes, and can accelerate the diagnosis of rare disorders. In human medical applications, metabolomics profiling has been applied in several clinical fields including respiratory, neurologic and kidney diseases, diabetes and metabolic disorders [29,30]. Similarly, lipidomics profiling, a subset of metabolomics devoted to the qualitative and quantitative analysis of lipids, has emerged as a promising new tool for identifying alterations in lipid classes and finding new biomarker candidates [31]. The human lipidome reflects the inter-individual variation in lipid species, and has the possibility to deepen our understanding of fundamental biological processes and enables a broader insight into disease causes and progression.

In the clinical setting, lipid measurements have remained unaltered for the past 60 years. By analyzing total triglycerides (TG), total Chol, LDL-C and HDL-C, the total lipidomic profile is poorly described, leaving a wealth of information undiscovered. Therefore, the field of clinical lipidomics has become a steadily growing and promising field to address congenital and rare diseases [32,33,34,35]. According to the Web of Science, related publication output in the last decade increased more than 17-fold compared to the previous decade, thereby becoming one of the fastest growing research fields. This was also reflected by a PubMed search using the keyword “Lipidomics”, which led to one hit in the year 2001, whereas, in the year 2021, over 1800 articles were associated with “Lipidomics” (Figure 2). Strikingly, combining the key words “Lipidomics” and “Rare Diseases” (equivalent to “Rare disorders”, “Orphan diseases”, “Orphan disorders”) in PubMed, only 22 articles were published in the year 2021, reflecting the underexploited potential of lipidomics in the field of rare diseases.

In the past few decades, mass spectrometry (MS) workflows have become more technically stable and user-friendly, paving the way to clinical routine laboratories and enabling simultaneous analysis of several lipids beyond TG and Chol. Current lipidomics research offers the potential to identify over 1000 lipid species divided into dozens of lipid classes and subclasses in a high-throughput manner using a small sample volume. This creates the opportunity for studying the cellular metabolism by determining differences in specific lipid (sub-)classes and molecular species that reflect metabolic variation [36]. Improving MS technology now and in future offers the possibility to identify and quantify lipids in smaller sample volumes of the brain, as well as in other tissues, eventually reaching single-cell analysis. Promising aspects for the future include the combination of statistics, epidemiological modeling and machine learning for establishing predictive models in order to identify at-risk individuals and stratify risk in diagnosed patients [37].

### State of the Art in Lipidomics

The lipidome is responsive to genetic and environmental influences, such as gene mutations, disease states, lifestyle, diet, medication and even interaction with the gut microbiome [38]. From a technical perspective, MS is the most important underlying technology in lipidomics, either coupled to a separation technique, such as chromatography, or without any prior separation, which is then often referred to as direct infusion or shotgun lipidomics. The lipidomics workflow is divided into several distinct steps, including preanalytics (sampling, lipid extraction etc.), data acquisition and data analysis (Figure 3) [39]. The most important step in this workflow is sampling, because whatever is lost at sample collection cannot be retrieved later on, even by the best analytical methods. As a rule of thumb, it is always recommended to either immediately process or freeze samples, because otherwise lipid hydrolysis or lipid peroxidation might distort the composition of the lipidome. Particular precaution has to be taken when analyzing lipid classes, such as oxidized lipids or lyso-PL, as these are typical products of the abovementioned biochemical processes and show a very low natural abundance, which can easily result in completely inflated quantitative figures [40]. When proceeding to sample extraction, by far the most widely used method is liquid-liquid extraction, where the protocols of Folch [41], Bligh and Dyer [42] and MTBE [43] extraction work equally well; with Folch being better for non-polar lipids, while the latter two are better suited for polar lipids. Lipid extraction has a two-fold advantage: (i) the enrichment of lipid compounds in the organic phase and (ii) stripping the sample of any polar interferences.

Subsequent determination of a lipid extract by MS without prior chromatographic separation is called direct infusion or shotgun lipidomics (Figure 3). The platforms used for this step in the lipidomics workflow can operate either on low or high mass resolutions, with the latter providing the advantage of accurate mass for the determination of elemental compositions at very high certainty. In any case, it is essential that not only the intact masses of lipids are determined, but also that characteristic fragments of the compound under investigation are generated by collision-induced dissociation (CID). In this respect, low resolution platforms rely on triple quadrupole technology and identify lipids by their intact mass and their characteristic fragments [44,45], whereas high resolution platforms mostly rely on Orbitrap [46,47] and QqTOF [48] technology, and additionally provide elemental compositions of parent and fragment masses alike. Systems including chromatographic separation prior to MS are generally referred to by the term LC-MS or LC-MS/MS, and their main benefit over direct infusion systems is the reduction of mass spectral complexity and the introduction of retention time for increased identification certainty. The most widely used chromatographic techniques are reversed-phase HPLC and hydrophilic interaction liquid chromatography (HILIC) (Figure 3). While the first separation mechanism separates lipids by their hydrophobic fatty acid moieties, the latter separates lipids by their polar headgroups into lipid classes (e.g., PCs, SM etc.). The advantage of reversed phase chromatography is its inherent ability to separate molecular lipid species within each lipid class, and thus increase identification coverage [49], while the merits of HILIC are based on improved quantitation properties [50]. Coupling to low mass resolution triple quadrupole instrumentation results in a typical targeted lipidomics setup relying on retention time, and one mass transition from an intact lipid mass to a characteristic fragment is also known as single reaction monitoring (SRM) [50]. In such a setting, hundreds of lipids can be identified in a single mass chromatographic run [51]. High resolution instrumentation like Orbitrap or QqTOF, on the contrary, can be used for targeted as well as for non-targeted lipidomics approaches, because the instrument acquires full scan mass spectra of all ionized intact lipids [52,53,54]. In a further step, the instrument runs CID fragment spectra either in a targeted fashion on predefined lipids or in a non-targeted data-dependent fashion on the most intense signals in the previously acquired full scan spectrum. Particularly when entering the non-targeted lipidomics realm, high mass resolution is mandatory for the identification of unknown compounds.

In recent years, ion mobility spectrometry (IMS) has become increasingly popular in lipidomics, because it adds drift time as an additional identification criterion to MS systems (Figure 3), which is at least, to some extent, complementary to *m/z* [55,56,57]. Although *m/z* values and drift times generally correlate, because bigger ions at higher masses do have longer drift times, IMS is still very useful for the separation of isomeric structures, which cannot eventually be separated by mass spectrometry. Since the drift time of an ion in IMS depends on its cross-collisional section (CCS) and this is an invariable property of each lipid, this can be used as a unique identifier for each lipid. Taking advantage of this fact, several groups have started to develop lipid CCS databases in recent years by either experimental determination [58] or in silico generation of CCS values [59] for individual lipid species. From this perspective, it is a highly likely assumption that IMS will increasingly contribute to lipid identification within the next decade.

Another interesting emerging technique is mass spectrometry imaging (MSI), either by matrix-assisted laser desorption (MALDI) or, to a lesser degree, by secondary ion mass spectrometry (SIMS). In the case of MALDI imaging, tissues are cryo-dissected into slices of a few micrometer thickness, positioned on a MALDI target plate and coated by the MALDI matrix. Subsequently, the MALDI target is scanned in two dimensions at a pixel size of a few micrometers by the laser, resulting in a two-dimensional distribution of all the *m/z* values included in the mass spectral acquisition range. In this whole process, the homogeneous deposition of the MALDI matrix is the most critical step because matrix inhomogeneity may result in decreased spatial resolution and a distorted distribution of *m/z* values. In recent years, the high mass resolution and MS/MS spectra generation capabilities of Orbitrap analyzers have been successfully integrated with MSI [60], resulting in positive outcomes; e.g., in the determination of highly complex sulfoglycosphingolipids [61]. Additionally, a new MALDI-2 source design results in elevated ionization efficiencies for certain non-polar lipid classes [62].

The final step in any lipidomics workflow is data processing and data analysis (Figure 3) and, up until now, this has been the biggest bottleneck in lipidomics. The main obstacle in this respect is the high degree of isomerism inherent to lipid structures, which often results in mixed spectra generated by overlapping isomeric and isobaric compounds. However, despite these obstacles, great progress has been made within the last decade, resulting in some software packages which enormously alleviate the lives of scientists. Basically, there are two concepts of spectral interpretation which are used by any processing software: (i) similarity search algorithms [63,64] and (ii) rule-based decision sets. While similarity search algorithms work with experimentally obtained lipid fragment mass spectra deposited in a database, rule-based decision systems create fragment masses and intensities in silico, based on known fragmentation patterns of lipids [65,66]. Since the latter system is able to extract spectral information from overlapping mixed spectra to a much higher degree, it also results in higher positive predictive values (PPV), which can be regarded as a surrogate parameter for identification reliability. Particular precaution must be taken when it comes to the annotation of identified lipid molecular structures. Due to the inherently high degree of isomerism encountered in lipidomics, each elemental composition can putatively be generated by hundreds of different underlying lipid structures, which simply reflects the combinatorial power of possible fatty acyl compositions, and the variability of double bond positions and their configuration (cis, trans) within each unsaturated fatty acyl moiety. Since it is difficult, in a lipidomics experiment, to pin down each molecular lipid structure to the level of double bond location and configuration, a shorthand nomenclature reflecting the level of structural certainty is required. Such a unique shorthand nomenclature for lipid annotation has been continuously developed over the last decade [67,68] in conjunction with universally accepted reporting standards [69], which makes it much easier for readers, reviewers and editors alike, to assess the depth of identification level for each lipid species and the overall quality of a study. The golden rule in this respect is: only report what is experimentally proven and accordingly annotate it.

## 4. Lipid*omic*s in Rare Diseases

Due to the known connection of altered lipid metabolism in rare diseases, lipidomics analysis provides a promising tool for the identification of several inborn errors of metabolism. Lipidomic changes in rare and undiagnosed diseases are often minor, consisting of complex patterns of subtle changes of a distinct set of lipids, which can be easily identified by lipidomics analysis [70]. The aim of clinical routine should be to expand from single lipid analysis to multianalyte lipid panels or lipidomic analysis in order to aim for more specific readouts concerning lipid-associated pathophysiologies with a potential application in personalized and “precision medicine”. In order to achieve these aims, interdisciplinary collaborations including routine clinics should be initialized in order to obtain workflows for clinical adaptation [71]. The NIH Common Fund Undiagnosed Disease Network (UDN) started an initiative to perform metabolomics and lipidomics analysis of 148 patients and family members in biofluids (urine, CSF, blood plasma) and make the raw and processed data available to the research community. The overall aim is to accelerate diagnosis and clinical management, advancing the research of rare and currently unrecognized diseases [70].

### 4.1. Genome-Wide Association Studies (GWAS)

Genome-wide association studies are observational studies of collected genomic information on genetic variants aiming to associate any variant with a feature. Hicks et al. performed GWAS using targeted lipidomics of 4400 subjects from five diverse European populations to assess the association of 318,237 single-nucleotide polymorphisms (SNPs) with levels of circulating SM, ceramides and glucosylceramides (GluCer) [72]. They observed strong association in or near seven genes functionally involved in ceramide biosynthesis and trafficking, with an additional 70 variants across 23 candidate genes involved in sphingolipid-metabolizing pathways. A focused lipidome GWAS in 2181 Finnish individuals identified a higher heritability in PCs with a high degree of unsaturation than PCs with low degrees of unsaturation [73]. Lipidomics-based GWAS (including 355 lipid species) of 650 individuals from the Amish founder population identified lipid species associated with two rare-population but Amish-enriched lipid variants [74]. They also identified associations for three rare-population Amish-enriched loci with several SL. In addition, a combined approach of whole-exome sequencing (WES) and SL profiling by MS identified DEGS1 as a disease-causing gene leading to a heritable SL disorder with hypomyelination and degeneration of both the CNS and PNS [12]. DEGS codes for the enzyme Δ4-dihydroceramide desaturase are responsible for the last step of ceramide biosynthesis. Mutations in DEGS1 affect de novo SL synthesis, leading to a changed dihydro-SL/SL ratio and the formation of the potentially neurotoxic sphingosine.

### 4.2. Neurodevelopmental Disorders

#### 4.2.1. Lysosomal Storage Diseases (LSDs)

LSD are characterized by mutations in genes encoding for various lysosomal proteins and enzymes leading to lysosomal dysfunction in a group of over 70 diseases. It was shown more than 50 years ago that LSD are associated with an altered lipid profile; e.g., increase of brain SM in Niemann-Pick disease and specific ganglioside accumulation in GM1- and GM2- (Tay Sachs) gangliosidosis [75]. In addition, levels of the glycosphingolipids sulfatide and lysosulfatide have been linked to the severity of neuropathy in metachromatic leukodystrophy [76,77].

Neuronal ceroid lipofuscinoses (NCL) are caused by mutations in fourteen known genes (CLN1-14) [78], not linked to one pathway and are the most common inherited progressive encephalopathies of childhood, with a prevalence of 1 in ~13,000 live births [79]. NCL are characterized by progressive neuronal death in the CNS, leading to epilepsy and mental and physical decline. Previously performed lipidomics analyses during post-mortem autopsies from humans and animals (e.g., sheep) revealed a changed lipid profile, including Chol, cerebrosides (galactosylceramide), SL, PL, poly-unsaturated FA (PUFA) and dolichols [80,81,82,83]. Dolichols are major lipid components of neuromelanin, a dark brown pigment present at high concentrations in dopaminergic neurons of the human substantia negra [84]. It was shown that dolichol phosphates were increased in infantile, late infantile and juvenile forms of CLN, whereas non-phosphorylated dolichols were markedly increased in the cerebral cortex of children with late infantile and juvenile types of CLN. Dolichol phosphates are also increased in the brains of GM1-gangliosidosis and Tay-Sachs disease patients [83]. In post-mortem autopsies on the cerebral tissue of children who had died from infantile and late-infantile (CLN1 and CLN8) forms of NCL, quantitative lipidomics analysis revealed significant alterations in the PL and SL pattern, particularly in PUFA [76,85]. Similarly, fibroblasts from patients with late-infantile NCL (CLN2) showed alterations in the FA composition of PL, including PUFA [86]. A biomarker would be extremely helpful to establish an early diagnosis in CLN2, as the early stage is uncharacteristic with single seizures and mild developmental delay, but has the option of enzyme replacement therapy (ERT) by an intraventricular device [87].

Sphingolipidosis Gaucher disease (GD) is a rare recessively inherited disorder caused by biallelic mutations in GBA1, leading to the deficiency of β-glucocerebrosidase, an enzyme responsible for the degradation of GluCer, a monohexosylceramide (MHC), inside lysosomes [88,89]. Due to the accumulation of GluCer and its inflammatory metabolite glucosylsphingosine (GluSph) in the lysosomes, abnormally outsized so-called ‘Gaucher cells’ are formed in the spleen and liver, which lead to inflammatory and neurologic manifestations [90,91]. GD is divided into three different clinical types. Type I is a visceral type and causes no neurological manifestations, albeit with markedly increased risk of Parkinson’s disease and Lew body dementia in adulthood [89,92]. Type II is the most serious form, including severe neurodegenerative symptoms early in life and a high mortality rate before adulthood. Type III develops during adulthood and displays a high prevalence, especially in the Ashkenazi Jewish population (1/111,111 worldwide; 1/855 Jewish population) [93]. Types II and III are often present in the Middle East and Asia. To date, the lipid Glucosylphingosine (Lyso-GB1) is known as the most reliable biomarker available for the diagnostic prognosis and disease/treatment monitoring of GD [94]. It meets the criteria as a biomarker, as it is easily accessible and quantifiable in plasma, as well as in dried blood spots. Treatment of non-neuronopathic GD1 involves ERT-targeting macrophages and substrate reduction therapies (SRT) using inhibitors of glucosylceramide synthase (GCS) [95,96]. Thus, massive lipid accumulation in the lysosomes is hampered by lowering excessive GluCer and, therefore, prevents further organ damage. Since GD is related to an altered lipid metabolism, resulting in MHC accumulation and an increase in PC and SM, Byeon et al. performed lipidomics analysis in plasma and urine samples of patients with GD during ERT and compared it to healthy controls [97]. In detail, GD patients showed an increase in various PL and SL, such as Ceramides, including MHC and dihexosylceramides (DHC), which was reversed during ERT, supporting lipidomics as a suitable monitoring technique for ERT. As neuronal injury and cell death are prominent pathological features in neuronopathic GD, Boddupalli et al. concentrated on the involvement of GBA in neuroinflammation driven by microglia activation [91]. In order to assess the impact of glycosphingolipid accumulation in various cell types of the brain, the efficacy of brain-permeable GCS inhibitors and the identification of new biomarker candidates, lipidomics analysis was performed. In detail, the targeted rescue of GBA in microglia and neurons, respectively, and the administration of GCS inhibitors reversed the accumulation of GluCer and GluSph, respectively, in GD model mice, concomitant with a reduction in neuroinflammation and improved survival.

Fabry disease is a multisystemic X-linked LSD, leading to the accumulation of glycosphingolipids, predominantly globotriaosylceramide (Gb3), in biological fluids and multiple organs and tissues, such as small vessel walls, unmyelinated nerves, the heart and kidneys. As Fabry disease is associated with a variety of symptoms, patients are often misdiagnosed or belatedly diagnosed. Gb3 and lyso-Gb3 have been applied as biomarkers in various biological fluids of patients with the potential to significantly improve Fabry patient’s diagnosis and screening, especially in female patients [98].

Niemann-Pick C disease (NPC) is a lysosomal storage disease, caused by mutations in NPC1 and NPC2 with a wide onset age, ranging from neonates to adults. Clinical symptoms are highly heterogenous, including neonatal jaundice and/or cognitive dysfunction, hepatosplenomegaly, dysarthia, dysphagia, verticular supranuclear gaze palsy, epilepsy and psychiatric manifestations. The disease incidence is estimated at 1 in 100,000 births, but could be underestimated due to late onset NPC1 phenotypes, which increase the frequency to 1:19,000–1:36,000 [99]. Classical NPC biomarkers are oxysterols (TRIOL, 7-KC) and lyso-SL (Lyso-SM, LysoSM-509). Boenzi et al. performed a quantitative lipidomics analysis of the plasma of 15 patients with NPC and age-matched controls [99]. In detail, the authors performed LC-MS/MS using Ion Mobility MS, allowing the simultaneous quantitation of ~1100 lipid species. In addition to confirming increased levels of oxysterols, lipid profiles of NPC patients compared to healthy controls showed elevated levels of total diglycerides (DG) and arachidonic acid, whereas levels of SM, PE, PC, CE and lactosylceramides were decreased. Importantly, levels of arachidonic acid revealed a positive correlation with the biomarkers Triol and LysoSm-509, indicating novel biomarkers for NPC.

The ultra-rare Mucopolysaccharidosis plus syndrome (MPS-PS) is caused by homozygous missense mutations in conserved regions of the VPS33A gene, a core component of the class C core vacuole/endosome tethering (CORVET) and homotypic fusion and protein sorting (HOPS) complexes, which play essential roles in endocytosis [100]. Patients with MPS-PS suffer from a broad spectrum of clinical symptoms, as seen in MPS, as well as other features, such as congenital heart defects, renal and hematopoietic manifestations leading to early death within 2 years of age [101]. MPS-PS is especially present in Yakuts, a nomadic ethnic group in Southern Siberia. Lipidomics screening displayed SL abnormalities in fibroblasts of patients with MPS plus syndrome [102]. In addition, concentrations of psychosine and the de-acylated form of galactosylcermaide are known to be increased with Krabbe disease.

Krabbe disease, also known as globoid cell leukodystrophy (GLD), is caused by mutations of galactosylceramidase (GALC), an enzyme responsible for hydrolyzing galactolipids, including galactosylceramide and psychosine (galactosylsphingosine). GLD is characterized by severe myelin loss, a reduction of oligodendrocytes, astrocytic gliosis, glial cell infiltration and rapid progression of nervous system dysfunction. To date, psychosine is the only lipid species that has been reported to be differently regulated in the brain of GLD patients [103]. In addition, it has been shown that its concentration rises with progression of the disease and severity [104]. In contrast to human samples, the naturally occurring canine GLD model opened up the opportunity to perform a longitudinal targeted lipidomics analysis of CSF and brain samples compared to age-matched controls. Corado et al. identified significant elevation of four lipid classes: Galactosylphingosine, Galactosylceramide, Glucosylsphingosine and Dihexosylceramide [104]. These changes were detectable at early disease onset and correlated with disease progression. Importantly, these lipids elevated in the CSF were also increased in the brain at endpoint, indicating that CSF samples serve as good indicators of CNS disease progression. In addition, this study showed that multiple lipids could serve as monitoring biomarkers for future clinical trials in GLD patients.

#### 4.2.2. X-Linked Disorders

X-Linked intellectual disability (XLID) is responsible for 5–10% of mental disabilities in men, including over 150 known inherited mental disabilities [105]. Even though XLID primarily affects men, a subset of X-chromosome-associated diseases, including fragile X, Rett (RTT), Hunter, Turner, and CDKL5 syndromes, also affect females. Yazd et al. found a distinct metabolomics and lipidomics profile of neural progenitor cells (NP) of a representative patient associated with X-chromosomal deletion disorder compared to a healthy control [106]. In detail, the results revealed perturbations in several metabolic pathways, including neurotransmitter biosynthesis and overall brain function and a lipid dysregulation, including a disturbed cellular structure and membrane integrity (VLCFA PPC highly expressed in XLID, altered PC/PE ratio).

Rett syndrome (RTT) is a rare X-linked dominant neurological disease caused by mutations in the methyl-CpG binding protein 2 (MECP2). RTT is one of the most common causes of genetic mental retardation in girls, characterized by normal infantile psychomotor development, followed by severe neurologic regression. RS lacks a specific biomarker, but altered Chol and lipid metabolism has been found in a KO-mouse model, as well as in RS patients. Our group performed both targeted and untargeted LC-MS/MS metabolomics and lipidomics to investigate the CSF and plasma composition of patients with RS for biochemical variations compared to healthy controls [107]. We were able to show that patients in our study cohort had decreased Chol levels in the CSF compared to the controls. In addition, levels of various PL and SM species were also decreased in the CSF of RTT patients. Plasma samples showed reduced levels of PL, whereas TG species were increased compared to the healthy controls.

The X-linked adrenoleukodystrophy (ALD) is caused by mutations in the ABCD1 gene leading to the accumulation of very-long-chain FA (VLCFA) [108]. In men, the symptoms include progressive spinal cord disease, primary adrenal insufficiency and cerebral inflammatory disease. Previously, women were considered to be asymptomatic carriers, even though more than 80% develop progressive spinal cord disease. The most important biomarker VLCFA, specifically the ratio FA26:0/FA22:0 and the ratio FA24:0/FA22:0, shows a sensitivity of almost 100% in men, whereas it detects only 15–20% in female carriers. Recently, it was shown that Lyso-PC 26:0 serves as a better diagnostic biomarker in female carriers than FA26:0, with an elevation in all DBS samples from ALD women. In a follow-up study, the authors concluded that progression of spinal cord disease cannot be detected with common diagnostic assessments, such as EDSS (Expanded Disability Statue Scale), ALDS (AMC Linear Disability Scale) and SF-36 in women after an 8-year follow-up period. New potential diagnostic biomarkers could be identified with a semi-targeted lipidomics approach [109].

Kallmann Syndrom (KS) is another X-linked rare genetic disorder characterized by hypogonadotropic hypogonadism accompanied by hyposomia or anosmia, which is caused by congenital gonadotropin-releasing hormone (GnRH) deficiency [110]. Lipidomics profiling of seminal plasma from patients with KS revealed decreased TG, PC and PE lipid species, indicating promising biomarkers for KS diagnosis.

One of the most common X-linked neuromuscular disorders, Duchenne Muscular dystrophy (DMD), is caused by mutations in the dystrophin-encoding gene DMD and is inherited in a recessive X-linked manner. DMD affects 1 in 5000 mostly male births, leading to the most common form of muscular dystrophy [111]. Due to the disrupted synthesis of dystrophin, patients with DMD suffer of delayed motor development, progressive muscle weakness, followed by wheelchair dependency and premature death. Previous studies have shown a changed FA composition of PC in the dystrophic muscle of DMD model mice compared to healthy muscle [112]. In line with this, affected muscles in DMD patients show alterations of PC, including higher levels of FA 18:1 chains and lower levels of FA 18:2 chains, possibly reflected in high levels of PC 34:1 and low levels of PC 34:2 [113]. Furthermore, MS-based lipidomics imaging showed that several compounds belonging to PC, Lyso-PC, phosphatidic acid (PA), PS and SM classes, as well as TG, were increased, while PC was decreased in DMD muscles compared with the control muscles [114]. In addition, lipidomics analysis of the plasma from DMD model mice revealed a strong lipidomics signature in dystrophic mice related to disease progression and compared to healthy control mice, suggesting the investigation of lipid metabolism in DMD patients as well [115]. In detail, plasma of DMD model mice showed a significant elevation of glycerolipids, such as TG species, and PL, including ganglioside GM2, SM and ceramide species, cholesteryl oleate and cholesteryl arachidonate, compared to the controls.

#### 4.2.3. PI4KA

In 2011, a novel genetic error in metabolism caused by mutations in PI4KA was shown to result in a broad phenotypic spectrum ranging from severe global neurodevelopmental delay associated with severe hypomyelinating leukodystrophy to severe forms of spastic paraparesis [116]. It has been previously shown in animal models, such as yeast, flies, mice and zebra fish models, that downregulation of its expression leads to profound abnormalities in development [117,118,119,120]. The enzyme phosphatidylinositol 4 kinase A (PI4KA) catalyzes the phosphorylation of PI to PI(4)P, the first and most important step in the phosphoinositide metabolism. Phosphoinositides, located in the plasma membrane, are important lipids in the brain, due to their responsibility in cell signaling and ion channel activity and membrane trafficking. By performing targeted lipidomics, the authors were able to confirm a significantly decreased PIP/PI ratio, indicating reduced PI4KA activity [116].

#### 4.2.4. POEMS Syndrome

Polyneuropathy, organomegaly, endocrinopathy, monoclonal protein and skin changes (POEMS) syndrome are rare disorders defined by monoclonal plasma cell disorder, peripheral neuropathy and other systemic symptoms. The pathogenesis of POEMS syndrome is poorly understood, but the overproduction of vascular endothelial growth factor (VEGF) is regarded to be an important criterion for disease activity and clinical manifestation. Untargeted lipidomic screening in the serum of POEMS patients compared to the controls revealed a distinct serum lipid profile, including fatty acyl 17-oxo-20Z-hexacosenoic acid, PC(16:0/18:1) and sterol lipid 5b-pregnanediol, in patients compared to the controls [121].

#### 4.2.5. Moyamoya Arteriopathy

Moyamoya arteriopathy (MA) is known as a rare cerebrovascular disease associated with Ring Finger Protein 213 variants [122]. Patients suffer from recurrent ischemic and hemorrhagic strokes, severe neurological deficits and progressive physical disabilities. To date, the underlying pathophysiology is unknown. As lipids play a role in neo-vascularization/angiogenesis and inflammation, Dei Cas et al. performed an untargeted and targeted lipidomics approach in the plasma of MA patients compared to healthy controls. MA patients revealed especially lower lipid levels of plasma membrane lipids, such as PC, PI and alkyl-PC compared to healthy study subjects.

### 4.3. Pulmonary Alveolar Proteinosis (PAP)

PAP is an ultra-rare disease, caused by mutations in the genes for surfactant protein-B (SFTPB), surfactant protein-C (SFTPC) and member A3 of the TB-binding cassette family of transporters (ABCA3) leading to the accumulation of surfactant components in the alveolae, and thus, impairing gas exchange [123,124]. Surfactant is a complex mixture of lipids and proteins, which coat the alveolar space [125]. Lipidomics analysis of BAL (Bronchoscopy and Bronchoalveolar Lavage) samples from patients with PAP revealed a massive increase in Chol (60-fold), cholesterol ester (23-fold), PE, PC, ceramides (130-fold) and other SL. Particularly, very long-chain ceramides, such as d18:1/20:0 and d18:1/24:0 were increased, contributing to the proapoptotic environment observed in PAP.

### 4.4. Wilson Disease (WD)

WD is a rare autosomal recessive disease, caused by mutations in the copper-transporting P-type ATPase-encoding gene ATP7B responsible for the transport of copper into bile from hepatocytes [126]. Mutations in this gene lead to the accumulation of toxic copper in various tissues and organs, priming for chronic liver disease, CNS abnormalities and psychiatric disturbances. Oxylipins are defined as bioactive lipids derived from omega-3 and omega-6 PUFA via the cyclooxygenase (COX), cytochrome P450 and lipoxygenase pathways. It has been shown that oxylipins are deeply involved in the regulation of inflammatory processes by evoking anti-inflammatory and pro-resolving mechanisms [127]. Establishing the oxylipin profiles of WD and healthy controls, using the patient’s plasma, revealed an increase in eight oxylipins in WD compared to the controls, indicating an involvement of oxidative stress damage, inflammation and peroxisome proliferator-activated receptor (PPAR) signaling pathways [126].

### 4.5. Retinal Diseases

Bietti crystalline corneoretinal dystrophy (BCD) is a rare autosomal recessive disorder, mainly affecting Asian populations, particularly Chinese and Japanese, caused by mutations in CYP4V2, a member of the cytochrome P450 family 4 involved in fatty acid metabolism [128]. BCD leads to the deposition of intraretinal crystals, leading to progressive night blindness and visual loss. Lipidomics analysis of the serum from twenty-two BCD patients and age-matched controls revealed a significant alteration of five lipid classes, including plasmalogens of PE (PPE), PI, GluCer, DG and TG, between BCD and healthy controls [129]. In addition, co-regulation between PPE and TG was markedly altered in BCD patients compared to healthy subjects, which may be pathologically relevant to BCD.

### 4.6. Ichtyosis

Ichtyosis are rare genetic keratinizing disorders, characterized by an impaired epidermal barrier and increased risk of microbial infection. Epidermolytic Ichtyosis (EI), Netherton syndrome (NS) lamellar ichtyosis and congenital ichtyosiform erythroderma (CIE) are known to show the highest prevalence. Lipidomics analysis of skin surface samples from ichtyosis patients and controls showed that the skin microbiome is markedly altered from healthy skin, and specific alterations predominate in various ichtyosis subtypes [130].

Sjörgen-Larsson syndrome is a rare neurometabolic syndrome, caused by the accumulation of fatty alcohols and aldehydes (FALDH) in plasma and skin due to a missing fatty aldehyde dehydrogenase (biallelic mutation in ALDH3A2) [131,132]. Patients suffer from intellectual disability, spastic paraplegia and ichtyosis. The underlying disease mechanisms causing the brain disorder are unknown. Magnetic Resonance spectroscopy showed abnormal lipid accumulation, but lipid profiles of patient’s brain are unknown. In a lipidomics approach using MS imaging, an accumulation of ether lipids as well as ether PL in both white and gray matter with a concomitant reduction in non-ether lipids has been detected. Especially, in white matter, the content of Ether PL is significantly increased compared to gray matter.

### 4.7. Mitochondrial Diseases

Mitochondrial diseases (MD) are caused by mutations in the nuclear or mitochondrial DNA, and particularly affect skeletal muscle, the brain and the heart, tissues and organs known for high-energy demands. MD are extremely challenging to treat, leading to high mortality rates, mostly in childhood [133]. Lipidomics analysis of plasma from twenty patients with MD (ten patients with mitochondrial encephalomyopathy with lactic acidosis-MELAS, three patients with Kearn-Sayre syndrome, one patient with chronic progressive external ophthalmoplegia-CPEO, myoclonic epilepsy with ragged red fibres-MERF) revealed a significantly altered lipid profile compared to the controls [134]. In detail, MD patients showed increased levels of medium- to long-chain FA acylcarnitines, which indicates an impairment of FA oxidation, resultant from the dysfunction of carnitine palmitoyltransferase 1 (CPT1), an enzyme involved in acylcarnitine synthesis via fatty acid transportation into the mitochondrial matrix. In addition, PI 38:6 and various PC species were increased, whereas lyso-PC species were decreased in MD patients.

Leigh syndrome is an inherited mitochondrial disease with a prevalence of 1 in 2000 births, and is especially present in the French-Canadian population in the region of Quebec (LSFC, Leigh Syndrome French Canadian); its origin are mutations in the leucine-rich pentatricopeptide repeat-containing protein (LRPRC) [135]. Due to impaired assembly of the oxidative phosphorylation (OXPHOS) machinery, patients with LSFC suffer from an early onset progressive neurodegenerative disorder, including necrotizing encephalopathy, metabolic and neurological crisis. Lipidomics analysis in the plasma of humans and in transgenic model mice was able to show lipid perturbations in mitochondria, as well as in peroxisomes with higher levels of acylcarnitines, and reduced levels of plasmalogens and docosahexaenoic acid [136].

#### Long-Chain Fatty-Acid Disorders (lc-FAOD)

Long-chain fatty acid oxidation disorders (lc-FAOD) are monogenetic inherited diseases affecting mitochondrial β-oxidation of FA with a chain length C > 12 [137]. Lc-FAOD are diagnosed during newborn screening by measuring the accumulation of specific acylcarnitines via tandem MS [108]. During phases of high-energy demand, the organism mostly relies on β-oxidation. Patients with lc-FAOD especially suffer during active phases of severe energy deficiency and accumulation of toxic metabolites, due to hampered entering of lc-FA to the β-oxidation. In detail, lc-FAOD include defects in carnitine palmitoyltransferase I and II (CPT I and II), very long-chain acyl-CoA dehydrogenase (VLCAD), long-chain 3-hydroxy-acyl-CoA dehydrogenase (LCHAD) and mitochondrial trifunctional protein (MTP). Treatment recommendations consist of a fat-restricted diet, including the application of medium-chain FA (MCT oil) and the prevention of phases of fasting. It has been recently shown in a murine model of VLCAD deficiency, that an alteration of the whole lipidome affects cellular function [138]. Lipidomics analysis of patient’s fibroblasts showed that monogenic diseases of lc-FAOD do not only affect FA degradation, but also lead to systemic alterations of the composition of complex lipids [137]. In addition, mitochondrial cardiolipins were remodeled concerning chain length and the degree of saturation. Moreover, the abnormal PC/PE ratio, the increased levels of plasmalogens and lyso-PL support the theory of inflammatory processes in lc-FAOD. Recently, Alatibi et al. showed that the application of saturated medium-chain FA (especially C7) leads to an altered composition of membrane lipids in patient’s fibroblasts, especially observed in LCHADD fibroblasts [133,139]. In addition, treatment with MCFA seemed to be particularly beneficial in fibroblast cell lines of FAOD patients by supporting mitochondrial metabolism and by enabling restoration of the SL metabolic flux, and reducing the protein expression known to be involved in neurodegenerative diseases. The study conducted in patient’s fibroblasts confirmed the positive therapeutic effects of MCFA and triheptanoin applied to lc-FAOD patients by energy supply and lipid remodeling.

## 5. Conclusions

Comprehensive and unbiased NGS techniques allow for the identification of causal genes and help in the elucidation of metabolic pathways. This provides the basis for investigations into the role of lipids in the pathogenesis of disease and potential drug targets. In addition, lipid species may serve as biomarkers for disease monitoring and treatment control. Artificial intelligence algorithms, even though still in the infancy stage, present an opportunity to combine big-data obtained from multi-omics approaches and further employ new modeling and statistical tools. This offers the great potential to identify (novel) rare inherited defects in a precise and timely manner, thereby paving the way to “precision medicine”.

## Figures and Tables

**Figure 1 ijms-24-01709-f001:**
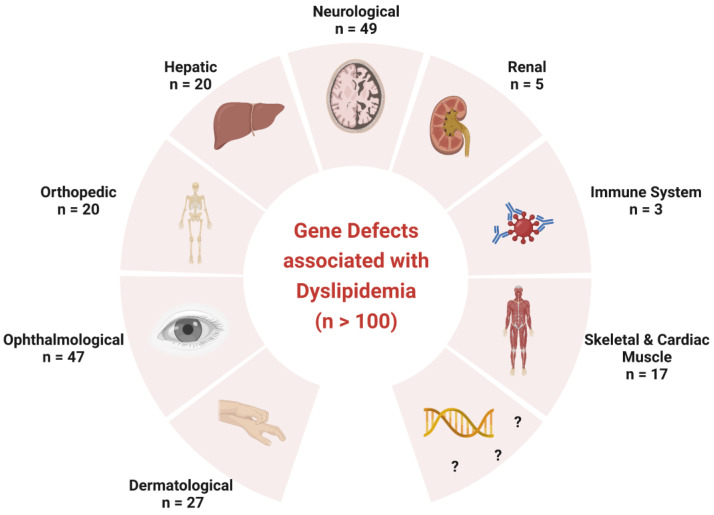
Congenital diseases associated with dyslipidemia to date. To date it is known that dyslipidemia is involved in more than 100 rare inherited metabolic diseases with multidisciplinary clinical manifestations including dermatological, ophthalmological, orthopedic, hepatic, neurological, renal, immune and skeletal presentations. The combination of present and future omics technologies will allow for the identification of new rare diseases. Figure created with BioRender.com.

**Figure 2 ijms-24-01709-f002:**
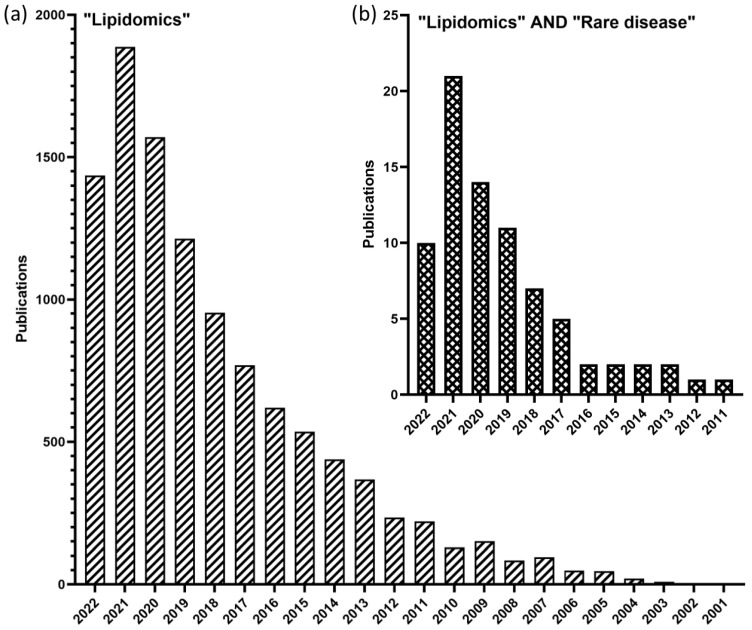
Publication output in the field of (**a**) “Lipidomics” alone, and (**b**) in combination with “Rare Diseases” according to PubMed. The keywords “Rare disorders”, “Orphan diseases” and “Orphan disorders” are equivalent to “Rare Diseases”, and have been considered in the evaluation. Date of PubMed search: 1 September 2022.

**Figure 3 ijms-24-01709-f003:**
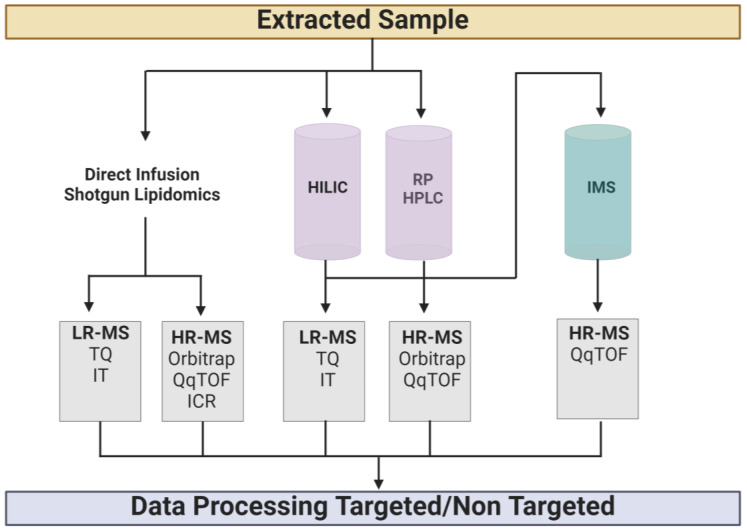
Mass spectrometric analysis workflow. HILIC, Hydrophilic interaction liquid chromatography; RP HPLC, Reversed Phase High Performance Liquid Chromatography; IMS, Ion Mobility Spectrometry; LR-MS, Low-Resolution Mass Spectrometry; HR-MS, High-Resolution MS; TQ, Triple Quadrupole; IT, Ion Trap; QqTOF, Quadrupole-Time-Of-Flight; ICR, Ion Cyclotron Resonance. Figure created with BioRender.com.

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
