# Peer review of "Lipidomics—Paving the Road towards Better Insight and Precision Medicine in Rare Metabolic Diseases"

_ijms, 2023, doi:10.3390/ijms24021709_

Round 1

Reviewer 1 Report

The authors provides a comprehensive review and indicates the importance of analyzing lipid profile for early detection of various disorders. Lipidomics can be a major pathway to precision medicine. Whereas the authors diligently covers the various neurological disorders and the role of lipid molecules in causing them, can they elaborate in the conclusion, how or which AI tools can be more helpful in analyzing the results obtained using LC/MS and the how these results can lead to precision medicine. There is typo in line 294 which the authors should correct -"precision medicine".

Reviewer 2 Report

The authors provided a comprehensive review of the roles of lipidomics in the diagnosis and prognosis of rare metabolic disease. It summarized the essential functions of lipids in disease pathologies and the rise of lipidomic research to facilitate researchers to study the mechanisms of rare metabolic disease. A detailed review of state-of-the-art lipidomic technology platforms involving mass spectrometry was presented and eventually, the applications of lipidomics in different types of rare diseases were categorized. This review is informative and would successfully raise people’s attention to lipidomics in the study of rare metabolic diseases.

The authors should address the following points before publication:

1. Numerous abbreviations were used in the manuscript without annotating the full term when they appear in the manuscript for the first time. i.e. Line 74 ‘APP’, line 88 ‘MS’ (multiple sclerosis), line 103 ‘PNS’, line 316 ‘WES’, line 350 ‘ERT’, line 634 ‘WGS’ - this one could be a typo for NGS.

2. The authors provided a relatively extensive review of the mass spectrometry based technology platform for lipidomics analysis. In section 4, lipidomics in rare diseases, MS imaging were mentioned multiple times (line 488, 577). However, this technology was not reviewed in section 3. It is necessary to add a paragraph to elaborate this.

3.   English needs more editing. I.e. line 122, line 625, line 191 - ‘like for example’ is not written language. Line 570 may need to be deleted.
